# c-Abl Phosphorylates MFN2 to Regulate Mitochondrial Morphology in Cells under Endoplasmic Reticulum and Oxidative Stress, Impacting Cell Survival and Neurodegeneration

**DOI:** 10.3390/antiox12112007

**Published:** 2023-11-16

**Authors:** Alexis Martinez, Cristian M. Lamaizon, Cristian Valls, Fabien Llambi, Nancy Leal, Patrick Fitzgerald, Cliff Guy, Marcin M. Kamiński, Nibaldo C. Inestrosa, Brigitte van Zundert, Gonzalo I. Cancino, Andrés E. Dulcey, Silvana Zanlungo, Juan J. Marugan, Claudio Hetz, Douglas R. Green, Alejandra R. Alvarez

**Affiliations:** 1Cell Signaling Laboratory, Department of Cell and Molecular Biology, Biological Sciences Faculty, Pontificia Universidad Católica de Chile, Santiago 8331150, Chile; 2Basal Center for Aging and Regeneration, Pontificia Universidad Católica de Chile (CARE UC), Santiago 8331150, Chile; 3Millennium Institute on Immunology and Immunotherapy, Biological Sciences Faculty, Pontificia Universidad Católica de Chile, Santiago 8331150, Chile; 4Department of Immunology, St. Jude Children’s Research Hospital, Memphis, TN 38105, USA; 5Center of Excellence in Biomedicine of Magallanes (CEBIMA), University of Magallanes, Punta Arenas 6210427, Chile; 6Institute of Biomedical Sciences, Faculty of Medicine & Faculty of Life Sciences, Universidad Andres Bello, Santiago 8370146, Chile; 7Department of Neurology, University of Massachusetts Chan Medical School (UMMS), Worcester, MA 01655, USA; 8Laboratory of Neurobiology, Department of Cell and Molecular Biology, Biological Sciences Faculty, Pontificia Universidad Católica de Chile, Santiago 8331150, Chile; 9Early Translation Branch, National Center for Advancing Translational Sciences (NCATS), NIH, 9800 Medical Center Drive, Rockville, MD 20850, USA; 10Department of Gastroenterology, Faculty of Medicine, Pontificia Universidad Católica de Chile, Av. Libertador Bernardo O’Higgins 340, Santiago 8331150, Chile; 11Biomedical Neuroscience Institute (BNI), Faculty of Medicine, University of Chile, Santiago 8330015, Chile; 12Center for Geroscience, Brain Health and Metabolism (GERO), Santiago 8380453, Chile; 13Program of Cellular and Molecular Biology, Institute of Biomedical Sciences, University of Chile, Santiago 8330015, Chile; 14The Buck Institute for Research in Aging, Novato, CA 94945, USA

**Keywords:** c-Abl, mitofusin 2, apoptosis, mitochondrial fusion, amyotrophic lateral sclerosis, endoplasmic reticulum stress

## Abstract

The endoplasmic reticulum is a subcellular organelle key in the control of synthesis, folding, and sorting of proteins. Under endoplasmic reticulum stress, an adaptative unfolded protein response is activated; however, if this activation is prolonged, cells can undergo cell death, in part due to oxidative stress and mitochondrial fragmentation. Here, we report that endoplasmic reticulum stress activates c-Abl tyrosine kinase, inducing its translocation to mitochondria. We found that endoplasmic reticulum stress-activated c-Abl interacts with and phosphorylates the mitochondrial fusion protein MFN2, resulting in mitochondrial fragmentation and apoptosis. Moreover, the pharmacological or genetic inhibition of c-Abl prevents MFN2 phosphorylation, mitochondrial fragmentation, and apoptosis in cells under endoplasmic reticulum stress. Finally, in the amyotrophic lateral sclerosis mouse model, where endoplasmic reticulum and oxidative stress has been linked to neuronal cell death, we demonstrated that the administration of c-Abl inhibitor neurotinib delays the onset of symptoms. Our results uncovered a function of c-Abl in the crosstalk between endoplasmic reticulum stress and mitochondrial dynamics via MFN2 phosphorylation.

## 1. Introduction

The endoplasmic reticulum (ER) is a membranous organelle key in the control of synthesis, folding, and sorting of proteins. The folding capacity of the ER is constantly challenged by physiological demands and disease states. To adjust proteostasis, cells engage a dynamic intracellular signaling pathway known as the unfolded protein response (UPR), enforcing adaptive programs that improve central aspects of the entire secretory pathway, whereas uncompensated ER stress results in oxidative stress and apoptosis [1]. The execution of cell death by ER-damaging insults largely depends on the intrinsic mitochondrial apoptosis pathway. Thus, a crosstalk between the ER and mitochondria is essential to determine cell fate under ER stress [2]. Interestingly, early studies suggested that c-Abl kinase operates as a signaling interphase between the ER stress and oxidative stress and mitochondria, mediated by its translocation to the mitochondria and the engagement of apoptosis programs [3]. However, the mechanisms and molecular targets of c-Abl at the mitochondria are not completely explored.

c-Abl is a non-receptor tyrosine kinase with functions in neurulation, cytoskeleton dynamics, synapsis, and apoptosis in the central nervous system [4,5,6,7,8,9,10,11,12,13,14]. Emerging evidence suggests that the localization of c-Abl to the cytosol has a significant impact on cytoskeleton dynamics, in addition to influencing cells under stress. For example, c-Abl is activated by oxidative stress impacting the activation of the canonical mitochondrial apoptosis pathway [15,16,17,18]. Also, c-Abl activation under cellular stress leads to the activation of the pro-apoptotic transcription factors p73 [19] and MST1 [20], and thus promoting the expression of proapoptotic genes. Several studies using the pharmacological targeting of c-Abl indicate that its activation has a pathogenic role in different human diseases linked to abnormal protein aggregation and oxidative stress. c-Abl exhibits a proapoptotic function in neurons exposed to amyloid β [21,22], and it is involved in neurodegeneration in Parkinson’s disease [23,24,25,26,27,28,29,30,31] and Alzheimer’s disease (AD) [32,33,34,35,36], in addition to lysosomal storage disorders [37,38] and amyotrophic lateral sclerosis (ALS) [39,40,41]. However, how c-Abl regulates mitochondrial dynamics and cell fate in cells under ER stress is still unknown.

Here, we report that c-Abl regulates mitochondrial morphology in response to ER stress. In cells under ER stress, c-Abl translocates to the mitochondria and triggers mitochondrial fragmentation. Interestingly, when c-Abl is pharmacologically or genetically inhibited in cells under ER stress, mitochondrial fragmentation and apoptosis are rescued. Then, we demonstrate that c-Abl phosphorylates mitofusin 2 (MFN2), an essential regulator of the mitochondria fusion machinery, specifically at Y269, which alters the MFN2 GTP binding affinity, influencing mitochondrial dynamics and cell survival under ER stress. Interestingly, the regulation of mitochondrial morphology by c-Abl-MFN2 has therapeutical potential, as demonstrated by its pharmacological inhibition using cell culture models for ALS. Furthermore, the treatment with neurotinib, a novel allosteric c-Abl inhibitor with high brain penetrance in ALS transgenic mice mutant for superoxide 1 (SOD1), results in a delayed disease onset. Overall, our results uncover that c-Abl modulates mitochondrial dynamics and apoptosis in cells under ER stress via MFN2 phosphorylation.

## 2. Materials and Methods

### 2.1. Primary Culture of Rat Hippocampal Neurons

Rat hippocampal cultures were prepared as described previously with some modifications [22,42]. Hippocampi from Sprague Dawley rats on embryonic day 18 were removed, dissected free of meninges in Ca^2+^/Mg^2+^-free HBSS, and rinsed twice with HBSS by allowing the tissue to settle to the bottom of the tube. After the second wash, the tissue was resuspended in HBSS containing 0.25% trypsin, and incubated for 15 min at 37 °C. After three rinses with HBSS, the tissue was mechanically dissociated in plating medium (DMEM; Invitrogen, Waltham, MA, USA), supplemented with 10% horse serum (Invitrogen), 100 U/mL penicillin, and 100 μg/mL streptomycin by gentle passage through Pasteur pipettes. The dissociated hippocampal cells were seeded onto poly-L-lysine-coated six-well culture plates at a density of 7 × 10^5^ cells per well in plating medium. The cultures were maintained at 37 °C in 5% CO_2_ for 2 h before the plating medium was replaced with Neurobasal growth medium (Invitrogen) supplemented with B27 (Invitrogen), 2 mM L-glutamine, 100 U/mL penicillin, and 100 μg/mL streptomycin. On day 2, the cultured neurons were treated with AraC 2 μM for 24 h; this method resulted in cultures highly enriched in neurons (∼5% glia). For hippocampal cultures from *ABL1* conditional knockout mice (*ABL1-cKO*), homozygous c-Abl-floxed mice were kindly donated by Dr. AJ Koleske (Yale School of Medicine, USA) and bred in our animal facility. *ABL1-cKO* mice were bred from *ABL1^loxP^/ABL1^loxP^* and Nestin-Cre^+^, obtained from Jackson Labs. These mice have loxP sites upstream and downstream of exon 5 of the *Abl1* gene. This strain was originated and maintained on a mixed B6.129S4, C57BL/6 background, and did not display any gross physical or behavioral abnormalities. Genotyping was performed using a PCR-based screening to evaluate c-Abl ablation [11]. Male and female mice were housed on a 12/12 h light/dark cycle at 24 °C with ad libitum access to food and water.

### 2.2. Immunofluorescence

The hippocampal neurons or MEF cells were seeded onto poly-l-lysine-coated coverslips in 24-well culture plates at a density of 2.5 × 10^4^ cells per well. The cells were rinsed twice in ice-cold PBS, fixed with a freshly prepared 4% paraformaldehyde/4% sucrose in PBS for 20 min, and permeabilized for 5 min with 0.2% Triton X-100 in PBS. After several rinses in ice-cold PBS, the cells were incubated in 3% BSA in PBS (blocking solution) for 60 min at room temperature, followed by an overnight incubation at 4 °C with primary antibodies. The cells were extensively washed with PBS and then incubated with Alexa-conjugated secondary antibodies (Invitrogen) for 60 min at room temperature. The cells were mounted in mounting medium and analyzed using confocal microscopy. The primary antibodies used were as follows: mouse anti-c-Abl, mouse anti-GAPDH, rabbit anti-actin, rabbit anti-βIII tubulin, and rabbit anti-TOM20 (SCBT, Dallas, TX, USA); rabbit anti-phospho-c-Abl Y412 (Sigma-Adrich, St. Louis, MO, USA); and cytochrome c (BD Biosciences, Franklin Lakes, NJ, USA). The mitochondrial marker Mitotracker Deep Red was obtained from Invitrogen.

### 2.3. Immunoblot Analysis

The treated cells were washed with ice-cold PBS and immediately lysed with radioimmunoprecipitation assay (RIPA) buffer (containing 50 mM Tris, 150 mM NaCl, 1 mM EGTA, 1 mM EDTA, 0.5% deoxycholate, 1% NP-40, and 0.1% SDS) supplemented with protease inhibitors (1 mM PMSF, 1 μg/mL aprotinin, 10 μg/mL leupeptin, 1 mM Na_3_VO_4_, and 50 mM NaF). The homogenates were maintained on ice for 30 min and then were centrifuged at 10,000× *g* for 5 min. The supernatant was recovered, and the protein concentration was determined using the BCA protein assay kit (Pierce, Appleton, WI, USA). The proteins were resolved in SDS-PAGE, transferred to a PVDF membrane, and reacted with primary antibodies. The reactions were followed by incubation with secondary peroxidase-labeled antibodies (Pierce) and developed using the ECL technique (Thermo Scientific, Waltham, MA, USA). The primary antibodies were the same as those used for immunofluorescence, with the addition of rabbit anti-p-CRKIII Y221, anti-caspase3, anti-cleaved-caspase 3, and anti-MFN2 (Cell Signaling Technology, Danvers, MA, USA), and rabbit anti-FLAG (Sigma Aldrich, St. Louis, MO, USA).

### 2.4. Coimmunoprecipitation Assay

Protein extract was obtained from MEF cells lysed in non-denaturing lysis buffer (20 mM Tris, 137 mM NaCl, 1 mM EDTA, and 1% NP-40) containing a mixture of protease and phosphatase inhibitors. Immunoprecipitations were performed using anti-c-Abl (SCBT), and MFN2 (Cell Signaling Technology) and agarose beads anti-FLAG (Sigma Aldrich). Complexes were isolated using protein G Sepharose. Tissue and cell lysates were separated by SDS-PAGE, transferred to Nitrocellulose membranes (Fisher Thermo Scientific), and immunoblotted with phosphotyrosine (pTyr) antibody (Millipore Bioscience Research Reagents, Burlington, MA, USA), anti-c-Abl, anti-FLAG, and anti-MFN2 antibody.

### 2.5. In Vitro Phosphorylation Assay

Kinase assay mixtures contained 25 mM HEPES, pH 7.25, 100 mM NaCl, 5 mM MgCl_2_, 5% glycerol, 100 ng of bovine serum albumin/μL, 1 mM sodium orthovanadate, and 10 nM c-Abl kinase. c-Abl was purified to >90% purity, as previously described [43]. After a 5 min preincubation at 30 °C, 25 μL reactions were initiated by the addition of immunoprecipitated agarose bead-FLAG-MFN2 or GST–CrkII, 5 μM ATP, and 0.25 μCi of [γ-^32^P]ATP. c-Abl incubated with 0.5 μCi of [γ-^32^P]ATP without a substrate was used for normalization. All reaction mixtures were incubated at 30 °C for different periods of time, and the reaction mixtures were washed four times with 0.75% *v*/*v* phosphoric acid. Alternatively, the reactions were terminated by the addition of ice-cold SDS sample buffer and were resolved by SDS-PAGE. The gels were dried and exposed for autoradiography, and quantified using a Molecular Dynamics PhosphorImaging system and the ImageQuant LAS 500 equipment.

### 2.6. Molecular Biology

pLZRS-FLAG-MFN2 was generated by PCR and restriction digest using 5′ EcoRI and 3′ XhoI. The pMSCV-mCherry vector has been described previously [44]. Outer mitochondrial membrane-targeted Cerulean (Cerulean-OMM) was generated by inserting the C-terminal domain of human BCL-xL (a.a. 208-233) downstream of Cerulean by PCR amplification with 5′ EcoRI and 3′ SalI restriction sites and cloning into pMX vector. The MFN2 mutant was generated by PCR using the proofreading Pfu polymerase (Promega, Madison, WI, USA), followed by DpnI digestion of the methylated parental plasmid. The oligonucleotides used were as follows: MFN2-Y269F-REV: 5′- GCACCTCCTCCATGAACTCAGGCTCCGA -3′; MFN2-Y269F-FOR: 5′- CTCGGAGCCTGAGTTCATGGAGGAGGTG -3′ (gene ID: 170731). pcDNA3-c-Abl-full-length, pcDNA3-c-Abl-kinase-dead-ful-length, and pcDNA3-c-Abl- ΔC (lacking the C-terminal domain a.a 1-634) were cloned into pMSCV-IRES-mCherry by PCR and restriction digest using 5′ Eco/kozak and 3′ SalI/STOP/ERT2-3′. All constructs were verified by DNA sequencing.

### 2.7. Cell Lines

MEF cells were obtained from ATCC. All stable MEF cell lines were generated by retroviral transduction. Briefly, Phoenix amphotropic virus producer cells were transfected with the appropriate plasmid using Lipofectamine 2000 (Thermo Fisher, Waltham, MA, USA) for 48 h. The target cells were infected with virus containing culture medium from the packaging cells supplemented with 5 μg/mL polybrene. Stable transductants were selected following the addition of 200 μg/mL Zeocin (Invitrogen, Waltham, MA, USA) (pLZRS vectors) or were sorted by flow cytometry for Venus-, mCherry-, or Cerulean-positive cells (pMX vectors).

### 2.8. Microscopy and Cell Death Assay

To assess the mitochondrial morphology, the cells were stained with Mitotracker Deep Red or with anti-TOM20. Images were collected using confocal microscopy (Zeiss LSM510, Thornwood, NY, USA) and analyzed with the NIH ImageJ 1.53 software. Fragmentation was judged based on the mitochondrial distribution of a normal cell, and cells exhibiting over 80% mitochondrial fragmentation were counted as fragmented. For each condition, at least 100 cells were counted. The quantification was carried out by an individual blinded to the conditions. TUNEL staining was performed using an apoptosis detection kit (Roche Molecular Biochemicals, Indianapolis, IN, USA). Briefly, the cells were incubated in 0.1% Triton X-100 in PBS. Then, the sections were immersed in the TUNEL reaction mixture for 60 min at 37 °C and washed twice in PBS (pH 7.4). Then, the cells were labeled with phalloidin-TRITC (red).

The confocal images of neurons were obtained using a Carl Zeiss 633 (numerical aperture 1.4) objective with sequential acquisition settings at the maximal resolution of the confocal (1024 × 1024 pixels), or using a Carl Zeiss Axiovert 200M motorized inverted microscope equipped with a precision motorized XY stage (Carl Zeiss MicroImaging, Thornwood, NY, USA). The confocal microscope settings were kept the same for all scans when fluorescence intensity was compared. All measurements were performed using the NIH ImageJ software.

### 2.9. Onset and Survival Analysis

Male WT or SOD1 G93A mice were fed ad libitum with a control diet, a diet containing nilotinib, or a diet containing neurotinib (a novel allosteric c-Abl inhibitor with high brain penetrance, patent number WO2019/173761 A1) [35,45]. The diet administration began from the time of weaning until euthanasia. The rodent diet was manufactured by Envigo/Teklad (Madison, WI, USA) with the incorporation of nilotinib at 200 ppm or neurotinib at 67 ppm into the NIH-31Open Formula Mouse/Rat Sterilizabile Diet (7017), followed by irradiation handling of the final product. All experimental protocols followed ethical guidelines. The onset was defined as the first day from birth when animals displayed weakness or coordination defects in their posterior limbs; i.e., when the animal was suspended by the tail, the hindlimb was collapsed, partially collapsed towards the lateral midline, trembled, or retracted. The endpoint was determined when animals were no longer able to feed themselves, or when the animal was suspended by the tail and there was rigid paralysis; or when the animal was allowed to walk and there was no forward motion; or when the animal was placed on its left and right side and it was not able to right itself within 10 s [46]. Euthanasia was performed with CO_2_, followed by perfusion with NaCl 0.9%. The lumbar spinal cord tissue was dissected, and half was frozen for Western blot analysis, while the other half was submerged in PFA 4% overnight for IHC analysis. On the next day, the spinal cord was transferred to 30% sucrose. For nerve tissue, sciatic nerves were dissected from each animal and fixed in 2.5% (*v/v*) glutaraldehyde in PBS for TEM analysis.

### 2.10. Fluorescent Immunohistochemistry

The lumbar spinal cord tissues, harvested from both WT and SOD1 G93A mice fed with either the control diet, nilotinib-containing diet, or neurotinib-containing diet, were frozen in the Tissue-Tek^®^ O.C.T. Compound (Sakura Finetek, Torrance, CA, USA) and subsequently sectioned using a cryostat at a thickness of 25 μm. The sections were collected every 200 μm in a solution of PBS with 0.02% azide.

To prepare the sections for analysis, the free-floating sections underwent permeabilization for 30 min using a 0.2% Triton X-100 solution in PBS, followed by blocking in a 3% BSA solution in PBS for 2 h at room temperature. Next, the sections were incubated overnight at 4 °C with a primary antibody against NeuN (Abcam, Cambridge, UK). On the following day, the sections were thoroughly rinsed with PBS, and then incubated for 2 h at room temperature and protected from light with the corresponding Alexa-conjugated secondary antibody (Invitrogen). This was followed by several washes with PBS. At least 6 sections were mounted on slides and allowed to dry overnight at room temperature, while being protected from light. On the following day, the slides were cover-slipped with glass slides using mounting medium.

Fluorescence images were captured using a Zeiss Axioscope 5 microscope. All images were consistently acquired using the same settings, and subsequently quantified using the NIH ImageJ software.

### 2.11. TEM and Morphological Analysis

The nerve tissue was prepared and fixed in 2.5% (*v/v*) glutaraldehyde in PBS. Ultrathin sections were mounted in a 300 mesh Formvar/carbon copper grids (Tedpella INC, Redding, CA, USA) and contrasted with uranyl acetate. Images were captured at different magnifications using a Philips Tecnai 12 transmission electron microscopy (Eindhoven, The Netherlands) at 80 kV equipped with a SIS CDD Megaview G2 camera and the iTEM Olympus Soft Imaging Solutions software (Windows NT 6.1).

The morphological analysis of TEM images was performed with the NIH ImageJ software. The images of individual mitochondria were generated from TEM images of sciatic nerve axons, and each mitochondrion was categorized as normal or swollen based on a disrupted external mitochondrial membrane with fragmented or swollen cristae/matrix, as described before [47].

### 2.12. Statistical Analysis

The mean and SEM values and the number of experiments are indicated in each figure. Statistical analysis was performed using one-way ANOVA, followed by Student’s *t* test using GraphPad Prism (version 5.0). For onset and survival analysis, a log-rank (Mantel–Cox) test was performed between curves. For IHC analysis, a one-way ANOVA with Dunnett’s T3 multiple comparison test was performed between groups. The results are presented as mean ± SEM.

## 3. Results

### 3.1. c-Abl Is Required to Induce Mitochondrial Fragmentation under ER Stress

To investigated whether c-Abl is involved in the morphological changes affecting mitochondria in response to ER stress in the nervous system, we evaluated if c-Abl was activated in neuronal cells exposed to the *N*-glycosylation inhibitor tunicamycin (1 µg/mL) or thapsigargin 1 μM. The primary hippocampal neurons exposed to tunicamycin or thapsigargin showed an increase in phospho-c-Abl signal on immunofluorescence assays (Figure 1A). This increase in phospho-c-Abl was also detected as early as 1 h after tunicamycin treatments, prior to and during ER stress, as shown by the accumulation of UPR markers CHOP and Bip by Western blot assays (Figure 1B). Then, we asked whether c-Abl tyrosine kinase is important for ER stress-induced mitochondrial fragmentation. The primary neuronal cultures under ER stress were treated with Imatinib (5 µM), a pharmacological c-Abl inhibitor. Immunofluorescence for βIII-tubulin (neuronal marker) and TOM20 (mitochondrial marker) showed that shortened mitochondria induced by ER stress were rescued by imatinib (Figure 1C,D), suggesting that activated c-Abl influences mitochondrial morphology. To confirm these results, we exposed MEF cells with tunicamycin and then we stained mitochondria with MitoTracker Deep Red FM to quantify mitochondrial sphericity in the cellular space using the IMARIS version 7.6.5. software. Higher values for sphericity are associated with more fragmented and/or rounded mitochondria. While the control and imatinib-treated cells exhibited a tubular mitochondrial network, tunicamycin treatment induced evident mitochondrial fragmentation, which was significantly prevented by imatinib (Figure 1E,F). These results suggest that the activation of c-Abl is required for the alterations of mitochondrial morphology elicited during the ER stress response. Importantly, the treatment of cells with imatinib protected against ER stress-induced cell death as demonstrated using the MTT (Figure 1G) and Sytox (Figure 1H) assays in the MEF cells. These effects correlated with the translocation of cytochrome c to the cytosol (Figure 1A) and the upregulation of ER stress-proapoptotic marker CHOP (Figure 1B).

Although imatinib is a well-known c-Abl inhibitor, other targets have been described such as PDGFR and c-KIT [48]. Thus, we employed a genetic approach to ablate c-Abl expression in the nervous system. We conditionally ablated c-Abl expression in the neurons by intercrossing *ABL1^flox/flox^* mice with Nestin-Cre transgenic mice to generate *ABL1-cKO*. First, we confirmed the successful deletion of c-Abl in embryonic brain extracts by Western blot (Figure 2A). We then generated primary neuronal cultures from *ABL1-cKO* mouse embryos and littermate control animals and evaluated mitochondrial length in neurites by staining for the mitochondrial marker TOM20 (Figure 2B). At basal level, *ABL1-cKO* neurons presented unaltered mitochondrial length (Figure 2C). Then, we analyzed the mitochondrial morphology of WT and *ABL1-cKO* neurons exposed to tunicamycin for 8 h. Although WT neurons exhibited a 50% reduction in mitochondrial length, *ABL1-cKO* neurons displayed mitochondrial lengths similar to control and untreated WT neurons (Figure 2C). These results indicate that c-Abl contributes to mitochondrial fragmentation under ER stress.

### 3.2. c-Abl Activation Induces Mitochondrial Fragmentation

To further explore the significance of c-Abl to mitochondrial dynamics, we performed gain-of-function experiments to study c-Abl activity in the absence of stress. To achieve this, we generated a c-Abl variant capable to activate its kinase function in a tamoxifen-inducible manner by fusing a C-terminal truncated c-Abl (constitutively active) to the ERT2 domain that blocks its kinase activity unless 4-hydroxy-tamoxifen (tamoxifen) is present (c-Abl WT ERT2). We also generated a kinase-dead variant (c-Abl KD ERT2) by substituting two specific amino acids (L285P/K290R) (Figure 3A). Then, to study the effect of c-Abl on mitochondria dynamics, we stably expressed both variants of c-Abl in plasmids carrying a mCherry reporter in MEF cells stably expressing the Cerulean mitochondrial marker [49]. As a control, we checked the activity of c-Abl by monitoring the phosphorylation of a well-known substrate CrkII (p-CrkII) at tyrosine 221 [50]. Hydrogen peroxide was also used as a positive control to activate endogenous c-Abl (Figure 3B). Tamoxifen-treated cells expressing the kinase-active c-Abl variant demonstrated a clear increase in the phosphorylation of CrkII, whereas the kinase-dead variant did not affect its phosphorylation (Figure 3B), demonstrating the specificity of our experimental system. We then analyzed the mitochondrial sphericity in MEF cells expressing the active or kinase-dead variants of c-Abl. We analyzed changes in tamoxifen-stimulated cells and compared mitochondrial sphericity with control cells carrying an empty vector (Figure 3C,D). Control cells exhibited tubular mitochondria (empirically minimal fragmentation), while FCCP treatment was used as a positive control to induce the maximal empirical mitochondrial fragmentation. Almost 80% of cells with stimulated c-Abl activity showed dramatic mitochondrial fragmentation, while the kinase-dead variant exhibited no significant fragmentation (around 25%), relative to control and FCCP (Figure 3C,D). These findings suggest that c-Abl activity is necessary for mitochondrial fragmentation.

### 3.3. c-Abl Phosphorylates the Mitochondria Fusion Protein MFN2 on Y269

To explore possible molecular mechanisms explaining the consequences of c-Abl activation on mitochondrial morphology, we investigated the potential interaction between c-Abl and components of the machinery controlling mitochondrial dynamics.

Using an in silico analysis based on the MitoCarta 3.0 dataset of 1136 mitochondrial proteins [51], we identified a group of proteins predicted to undergo phosphorylation by c-Abl (Appendix A and Table 1). Specifically, we identified a possible interaction between c-Abl and the outer mitochondrial membrane GTPase MFN2, which is essential for mitochondrial fusion [52,53,54]. We then tested whether c-Abl is capable to form a protein complex with MFN2. Endogenous c-Abl was immunoprecipitated from tunicamycin-treated MEF cells, and its association with MFN2 was assessed by Western blot analysis. Remarkably, an interaction between c-Abl and MFN2 was observed preferentially under ER stress (Figure 4A). Similar results were obtained when endogenous MFN2 was immunoprecipitated (Figure 4A). Interestingly, this interaction was reduced by the treatment of cells with imatinib (Figure 4A). We then stably reconstituted double-knockout MFN1/2 (MFN1/2 DKO) MEF cells with FLAG-tagged WT MFN2 (FLAG-MFN2) at levels that were similar to endogenous protein (Appendix A), and we induced ER stress in the presence or absence of imatinib. We detected increased c-Abl levels in FLAG immunoprecipitates after treating cells with tunicamycin, an interaction that was reduced after the administration of imatinib (Figure 4B). These results suggest that the activation of c-Abl is required to form a complex with MFN2 and in response to ER stress.

Although MFN2 has been reported to be phosphorylated at Ser27 in response to cellular stress [55], the phosphorylation in tyrosine residues of MFN2 remains unclear. To address this, we first examined whether MFN2 was phosphorylated at tyrosine in cells under ER stress. FLAG-MFN2 reconstituted in MFN1/2 KO cells expressing FLAG-MFN2 was treated with tunicamycin, and FLAG immunoprecipitated to assess phospho-tyrosine levels. Cells undergoing ER stress showed increased detection in tyrosine phosphorylation of MFN2, detected by the phospho-tyrosine antibody 4G10, and this was reduced in cells treated with imatinib (Figure 4C). The tyrosine phosphorylation of MFN2 was further evaluated in living cells by the addition of ^32^P-labeled orthophosphoric acid into the media of FLAG-MFN2-overexpressing MEFs using our tamoxifen-inducible system. Again, the expression of c-Abl kinase activity increased the radioactive signal of immunoprecipitated FLAG-MFN2 (Figure 4D), suggesting that MFN2 can be phosphorylated by active c-Abl in living cells.

Our in silico analysis provided the highest score for phosphorylation to the amino acid residue Y269 in the human peptide sequence 261-ASASEPEYMEEVRRQ-277 of MFN2 (Table 1), which is highly conserved in a wide range of species (Appendix A). Thus, we performed site-directed mutagenesis to generate a Y269F mutant. Both WT and Y269F FLAG-MFN2 versions were expressed at similar levels (Appendix A). We then treated these cells with tunicamycin and measured the levels of phosphorylated tyrosine in MFN2. As expected, phospho-tyrosine levels were detected in FLAG-immunoprecipitants from MFN2-WT-expressing cells but not in cells expressing the Y269F point mutant (Figure 4E). Then, to examine whether c-Abl directly phosphorylates MFN2, we performed an in vitro assay using purified recombinant proteins. Purified recombinant c-Abl kinase was incubated with FLAG-MFN2-WT, FLAG-MFN2-Y269F, or GST-CRKII (used as a positive target for c-Abl phosphorylation) in the presence of [γ-^32^P]ATP. We observed a significant time-dependent increase in FLAG-MFN2 phosphorylation measured via both autoradiography (Figure 4F) and scintillation counting (Figure 4G). The phosphorylation of FLAG-MFN2 WT was detected as early as 5 min of incubation and increased up to 120 min. In contrast, the phosphorylation of mutant FLAG-MFN2-Y269F remained at baseline levels until the endpoint. MFN2 fusion activity relies on the function of its GTPase domain [56]. In addition, it has been reported that the specific phosphorylation of MFN2 at Ser27 under cellular stress regulates mitochondrial fragmentation [55]. Using the Protein Imager tool [57], we modeled the Y269-containing GTPase domain of MFN2 (Figure 4H). This model predicted a close proximity of Y269 to the GTP-binding site. Based on this observation, we then addressed whether the Y269 residue of MFN2 affects its nucleotide-binding affinity upon ER stress using a GTP-agarose bead immunoprecipitation assay. While GTP binding by MFN2-WT was reduced upon ER stress, the Y269F mutant displayed a higher affinity for GTP, an activity that was unaffected by ER stress (Figure 4I). This suggests that MFN2-Y269F may remain in an active conformation for longer time than the WT form. We then measured the MFN2 hydrolytic activity and did not detect a significant difference in the GTP hydrolysis rate between MFN2 WT and the mutant form (Appendix A). Importantly, it is known that a change in the GTP hydrolysis rate is not necessary for mitochondria fusion and MFN2 activity [58]. Thus, the phosphorylation in Y269 appears to decrease the affinity of MFN2 for GTP, and this may reduce its ability to promote mitochondrial fusion. Altogether, these results indicate that MFN2 is directly phosphorylated by c-Abl kinase to control mitochondrial fragmentation.

### 3.4. c-Abl Mitochondria Localization Requires MFN2 Y269

We then investigated whether the interaction with MFN2 is necessary for c-Abl to localize in the mitochondria. To test this, we used FLAG-MFN2-WT and FLAG-MFN2-Y269F constructs on MEF cells, and then we determined the mitochondrial localization of c-Abl after tunicamycin treatment. Using stochastic optical reconstruction microscopy (STORM), we observed that the individual molecules of c-Abl were distributed diffusely with respect to mitochondria under basal conditions in cells expressing MFN2-WT or MFN2-Y269F (Figure 5A,B). Interestingly, ER stress induced a significant increase of c-Abl molecules and in large clusters at the mitochondria, a pattern that was absent from FLAG-MFN2-Y269F-expressing cells (Figure 5A,B). We then evaluated the proximity between c-Abl and MFN2 molecules after the induction of ER stress using STORM. In FLAG-MFN2-WT cells, the proximity within 50 nm was significantly increased after tunicamycin treatment. Importantly, this effect was ablated in FLAG-MFN2-Y269F cells (Figure 5C,D). Thus, c-Abl mitochondrial localization in response to ER stress requires MFN2 Y269.

Then, we studied whether the phosphorylation of MFN2 on Y269 by c-Abl is important to mitochondrial morphology. To analyze this, we used MEF cells that expressed FLAG-MFN2-WT or FLAG-MFN2-Y269F, and then stained them with TOM20 to study mitochondria morphology (Figure 6A). In control treatment, both cell lines exhibited normal tubular mitochondria morphology, whereas ER stress triggered significant fragmentation in 70% of FLAG-MFN2-WT-expressing cells but only in 45% of FLAG-MFN2-Y269F cells (Figure 6A,B), suggesting that the association between c-Abl and MFN2 is necessary to trigger mitochondria morphology impairments in cells under ER stress. Mitochondrial dynamics have the potential to influence mitochondrial function. In our observations, we found that ER stress did not affect the oxygen consumption rate (OCR) response (Appendix A). However, it did lead to a reduction in the extracellular acidification rate (ECAR) in MFN2 WT MEFs (Appendix A), indicating that the cells were still capable of oxygen consumption, but with altered glycolytic activity. In the case of the MFN Y269F mutant MEFs, treatment with tunicamycin did not produce changes in either OCR (Appendix A) or ECAR (Appendix A), suggesting that these cells could rely on both oxidative phosphorylation and glycolysis even when subjected to ER stress. Furthermore, our previous findings revealed that imatinib treatment protects against tunicamycin-induced cell death (Figure 1G). Similar results were observed using more specific apoptotic markers including cleaved caspase-3 both by Western blot (Figure 6C) and immunofluorescence (Figure 6D,E) and TUNEL assay (Figure 6F,G). Altogether, these results showed that the phosphorylation of MFN2 on Y269 by c-Abl is important to induce mitochondrial fragmentation and apoptosis in cells under ER stress.

### 3.5. c-Abl Tyrosine Kinase Inhibition Reduces Abnormal Mitochondria and Increases the Survival in an ALS Animal Model

ER stress has been extensively described as a pathological mechanism in neurodegenerative diseases associated with abnormal protein aggregation, including ALS [59]. Interestingly, c-Abl has been shown to contribute to the neurodegenerative cascade observed in several neurodegenerative diseases where ER and oxidative stress has been involved on the pathological causes [21,22,23,24,25,26,27,28,29,31,32,34,35,36,37,38,39,40,41]. To assess the significance of our findings to disease conditions affecting the nervous system, we explored the involvement of c-Abl in mitochondrial fragmentation in cellular and animal models of ALS. First, we expressed ALS-associated mutant SOD1-G85R or WT SOD1 fused to EGFP in primary hippocampal neurons at 7 days post differentiation. As expected, the expression of mutant SOD1 resulted in protein aggregation within hippocampal neurons, whereas the transfection of the WT SOD1 protein showed no such aggregation (Figure 7A). Also, neurons overexpressing mutant SOD1 exhibited higher levels of phosphorylated c-Abl at Y412 (Figure 7A), which was inhibited with imatinib (Figure 7A). We then analyzed mitochondrial fragmentation induced by mutant SOD1. Remarkably, imatinib significantly prevented mitochondrial fragmentation in neurons overexpressing mutant SOD1 compared with the WT form (Figure 7B,C).

Moreover, we evaluated the effect of pharmacological inhibition of c-Abl with nilotinib on the SOD1 G93A ALS mice. First, we quantified the percentage of morphologically disrupted mitochondria, characterized by an altered external mitochondrial membrane with fragmented or swollen cristae/matrix, in sciatic nerves from WT or SOD1 G93A mice fed with either control or nilotinib diet (Figure 7D). The results showed an increase of disrupted mitochondria in SOD1 G93A compared to control littermates. In addition, nilotinib treatment decreased the number of disrupted mitochondria in SOD1 G93A mice, while nilotinib did not influence the mitochondria of WT mice (Figure 7E). These results suggest that c-Abl inhibition with nilotinib may have a beneficial effect on altered mitochondrial morphology in SOD1 G93A mice.

To determine the protective effects of nilotinib in SOD1 G93A mice, we quantified the number of NeuN-positive neurons in the ventral horn of the lumbar spinal cord (Figure 7F). As previously described [60], we found a lower amount of NeuN cells in SOD1 G93A mice than in control littermates (Figure 7G). Interestingly, treatment with nilotinib prevented the NeuN-positive cell loss in SOD1 G93A mice, while control mice did not exhibit significant changes (Figure 7G).

When we analyzed the progression of the disease in control, nilotinib, or the recently described c-Abl inhibitor neurotinib- [35,45] fed SOD1 G93A mice, the results showed that neurotinib administration led to a significant delay in disease onset compared to the control group (*p* = 0.0081), whereas nilotinib did not exhibit a significant effect (*p* = 0.2253). The median onset of symptoms was 152 days in the neurotinib-treated mice, compared to 139 days in the control group (Appendix A). The median survival time was also slightly longer in the neurotinib group (165.5 days) compared to the control group (163 days), although this difference was not significant (*p* = 0.1237) (Appendix A). Overall, the results suggest that pharmacological c-Abl inhibition has a beneficial effect on the phenotype of the SOD1 G93A ALS mouse model, in delaying disease onset and improving neuronal survival.

## 4. Discussion

Mitochondrial dynamics influence the morphology and impact on various processes of mitochondrial homeostasis. In addition to the close relationship with the cellular bioenergetic capacity and buffering intracellular Ca^2+^, among other functions, mitochondria interact with the endoplasmic reticulum and react to cellular stress to favor apoptosis after severe injuries. Mitochondrial dynamics allow cells to adapt themselves to physiological and environmental fluctuations. Considering the high functionality of mitochondria, it is not unexpected that the machinery for fusion and fission follows rigorous regulations to keep cellular stability [56,61,62]. However, the mechanisms controlling the mitochondrial dynamic machinery are just partially understood. In this work, we unveiled a new molecular mechanism for c-Abl kinase in mitochondria as a modulator of the mitochondrial dynamics through MFN2 in response to cellular stress. We described that activated c-Abl promotes the mitochondrial fragmentation mediated through the phosphorylation of MFN2 at Y269 in response to ER stress.

The function for activated c-Abl under cellular stress has been mainly associated with the transcription of proapoptotic genes through the factors p73 and MST1 [20,21,22]. Additionally, it has been described that c-Abl is activated in response to ER stress, oxidative stress, and genotoxic stress [17,22,63,64], and translocates to the mitochondria where it promotes the outer mitochondrial membrane permeabilization and apoptosis [3]. However, the mechanism through which c-Abl could exert this function is unknown. Here, we focused on a potential mechanism for c-Abl in mitochondria. Given that ER stress leads to c-Abl activation, and it also affects mitochondrial dynamics [53], we investigated a potential role for c-Abl kinase in mitochondrial dynamics in response to ER stress.

In our first approach, we found that the mitochondrial fragmentation was reduced with imatinib, a specific c-Abl inhibitor, in response to ER stress, which agrees with previous studies in animal models concerning neurodegenerative disorders, where imatinib treatment had proven benefits, ameliorating symptoms [65]. Surprisingly, the expression of constitutively active c-Abl kinase by the tamoxifen-inducible system led to a dramatic mitochondrial fragmentation, while the expression of the kinase-dead variant did not induce changes on the morphology, demonstrating that activated c-Abl kinase is sufficient to induce mitochondrial fragmentation. Furthermore, c-Abl-deficient neurons exposed to tunicamycin revealed a significantly reduced mitochondrial fragmentation. Hence, using different strategies, our data indicates a novel function for c-Abl kinase collaborating in the mitochondrial fragmentation induced by cellular stress.

Through STORM microscopy, we provided detailed evidence for the c-Abl localization at mitochondria. In basal conditions, we found that c-Abl molecules were apparently randomly distributed, and interestingly, c-Abl molecules were dramatically enriched in mitochondria with a clustered-like pattern in response to ER stress. Through biochemical studies, it has been described that mitochondrial fractions are enriched in activated c-Abl under ER stress, and furthermore, c-Abl collaborates with the outer mitochondrial membrane permeabilization and apoptosis [3,66], supporting our results. In addition, it was reported that activated c-Abl exposes its N-terminal myristoyl group, which permits its localization with membranes [66] and could explain our observations of enriched c-Abl at mitochondria.

We explored potential targets of c-Abl that could be related to their association with mitochondrial dynamics, and found that c-Abl interacts with and phosphorylates the GTPase MFN2. Post-translational modifications on MFN2 have been previously reported in the literature. The phosphorylation of MFN2 at Ser27 by the JNK kinase was identified in response to cellular stress [55]; also, PINK1 phosphorylates MFN2 at Thr111 and Ser442, favoring its ubiquitination by the E3 ubiquitin ligase Parkin, in response to cellular stress [67]. Interestingly, those post-translational modifications reported for MFN2 are described as leading to mitochondrial fragmentation in response to cellular stress. Recently, it has been described that c-Src tyrosine kinase could regulate ER–mitochondrion interactions through the phosphorylation of the C-terminal tail of MFN2 [68]. Hence, our results, revealing the MFN2 phosphorylation by c-Abl kinase in response to ER stress, are consistent with this tendency, as described also with the phosphorylation of MFN1 by ERK in response to DNA damage [69,70].

A structural model prepared with the Protein Imager tool suggests that the Y269 amino acid residue is localized close to the GTP binding site and exposed to the cytoplasm, in the GTPase domain. Interestingly, the Y269 of MFN2 interacts with the N161 of the counterpart MFN2 monomer in the interface of the dimer. This interaction is crucial for mitochondrial elongation, as living cells seem quite sensitive to mutations in the interface [71]. These findings suggest that the Y269 residue plays a pivotal role in mediating protein–protein interactions within the MFN2 dimer interface, ultimately influencing mitochondrial fusion dynamics.

We found that the non-phosphorylable MFN2 Y269F exhibits an increased GTP affinity, which could suggest an augmented exchange from GDP to GTP. Additionally, we did not observe changes in the hydrolytic capacity of GTP between MFN2 WT and MFN2 Y269F, which could be in agreement with our data, since it was reported that changes in the hydrolytic capacity for MFN2 do not alter the mitochondrial fusion rates [58].

Through the STORM super-resolution microscopy, we detected that c-Abl colocalizes with MFN2 preferentially under stress, and interestingly, it was prevented with the expression of the mutant MFN2 Y269F. It seems that the Y269 residue of MFN2 is important to facilitate the c-Abl localization and enrichment in the mitochondrial membrane. It has been reported that MFN2 is enriched at mitochondria-associated membranes (MAMs) [72,73,74]. In this regard, whether c-Abl is localized at MAMs in response to ER stress remains to be determined.

We explored the biological significance of the mutant MFN2 Y269F to the mitochondrial morphology. Interestingly, the cells expressing MFN2 Y269F exhibited resistance to promote mitochondrial fragmentation in response to ER stress, suggesting that the phosphorylation of MFN2 at Y269 is required in the reduction in its pro-fusion function in response to ER stress. Our results suggest that the increased association with c-Abl could lead to the phosphorylation of MFN2 in response to ER stress. It has been reported that the phosphorylation of MFN2 at Ser27 is required to promote mitochondrial fragmentation in cells under genotoxic stress [55]. The use of other MFN2 mutants such as phosphomimetic MFN2 Y269E could be included in future experiments to explore more deeply the contribution on mitochondrial morphology; however, our approach confirms that the Y269 participates in the c-Abl–MNF2 interaction. Finally, we evaluated potential changes in the apoptotic response in MFN2 Y269F-expressing cells. Our findings show reduced levels for cleaved caspase 3 and TUNEL activation in MFN2 Y269F-expressing cells after an extended exposure to ER stress. During ER stress, c-Abl is activated, but its role in mitochondria remains unknown [3]. Our model reveals that c-Abl promotes a reduced function pro-fusion for MFN2, and it also favors the apoptotic response. Interestingly, post-translational modifications in MFN1 and MFN2 in response to cellular stress facilitate the apoptotic response [55,67,70], supporting the role for c-Abl promoting mitochondrial fragmentation mediated by its association with MFN2 in response to ER stress.

The activation of c-Abl in ALS motoneurons has been associated with oxidative stress [39]. Bosutinib, a Src/c-Abl inhibitor, reduced the misfolded mutant SOD1 protein levels, improved the expression of mitochondrial genes, and modestly extended G93A ALS mouse model survival [40]. Our results with nilotinib showed a coherent preservation of the mitochondrial morphology, and although we did not observe a significant increase in life span, the novel allosteric inhibitor neurotinib delays the onset of symptoms in the G93A mouse model, resembling the effects described for the c-Abl inhibitor dasatinib that also delays motor neuron degeneration in the G93A mouse model [41].

Overall, our study provides a view into the complex relationship between c-Abl kinase and mitochondrial dynamics mediated by MFN2 phosphorylation at Y269 during ER stress, mainly in the context of ALS. The correlation between in vitro and in vivo results reinforces the significance of c-Abl activity in mediating mitochondrial responses to stress conditions and the use of c-Abl inhibitors for targeting mitochondrial dysfunction in ALS.

## 5. Conclusions

In conclusion, our research highlights c-Abl kinase in regulating mitochondrial dynamic under cell stress. Specifically, we identified a new molecular mechanism by which c-Abl phosphorylates the mitochondrial fusion protein MFN2 at Y269 in response to ER stress, leading to mitochondrial fragmentation and apoptosis.

Our results point to a therapeutic potential of c-Abl modulation in neurodegenerative diseases like ALS, where ER and oxidative stress contribute to cell death. Moreover, the pharmacological inhibition of c-Abl in an ALS mouse model improves mitochondrial health and delays the onset of symptoms, opening avenues for further research and potential treatments in neurodegenerative conditions linked to cellular stress and mitochondrial dysfunction.

## Figures and Tables

**Figure 1 antioxidants-12-02007-f001:**
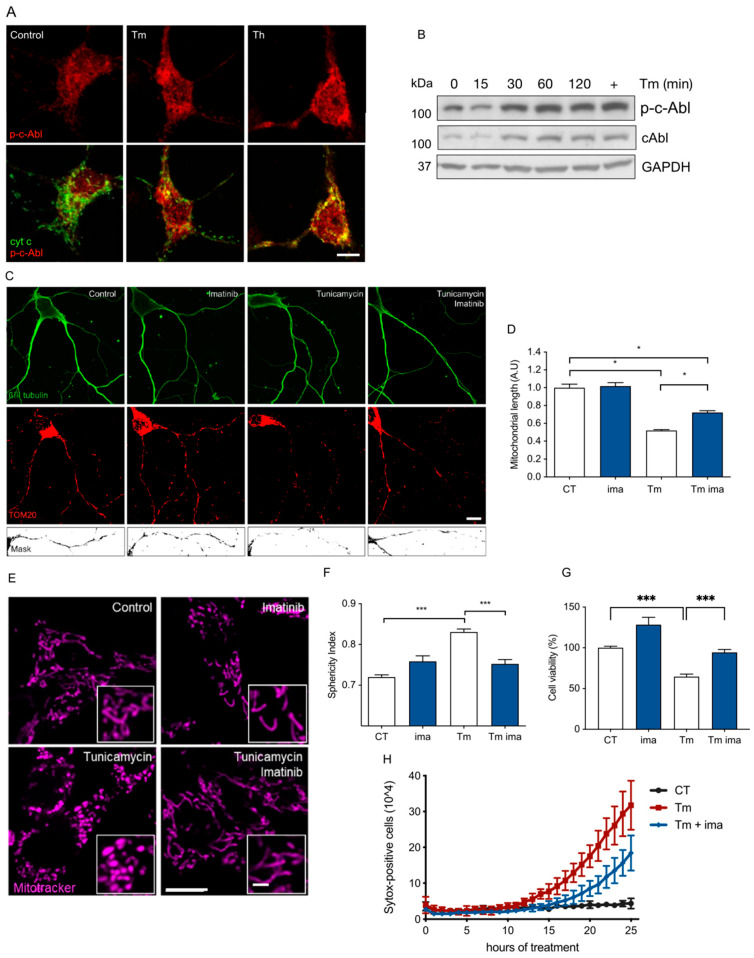
c-Abl is activated and localized in mitochondria in response to ER stress and collaborates in mitochondrial fragmentation. (**A**) Hippocampal primary neurons of 7 days in vitro were exposed to tunicamycin (Tm) or thapsigargin (Th) for 2 h and the immunodetection against phospho-c-Abl (red) was analyzed regarding mitochondrial localization with cytochrome c (green). Scale bar: 5 μm. (**B**) p-c-Abl and ER stress markers are increased after Tm treatment in hippocampal primary neurons. (**C**) Mitochondrial morphology of primary hippocampal neurons detecting TOM20 (red) and βIII tubulin (green) in treatments with Tm or Imatinib (Ima) for 10 h. Scale bar: 5 μm. (**D**) Quantification of mitochondrial length from neuronal processes in experiments performed in (**C**). (**E**) Mitochondrial morphology of MEFs detecting Mitotracker (magenta) in treatments with Tm or Ima for 10 h. Scale bar: 10 μm; inset: 5 μm. (**F**) Mitochondrial sphericity from experiments performed in (**E**). (**G**) MTT assay in MEFs in treatments with Tm or Ima for 10 h. (**H**) Sytox assay in MEFs in treatments with Tm or Ima for the indicated times. One-way ANOVA with Bonferroni post-test. Results are presented as mean ± SEM. *p* * < 0.05, *p* *** < 0.001.

**Figure 2 antioxidants-12-02007-f002:**
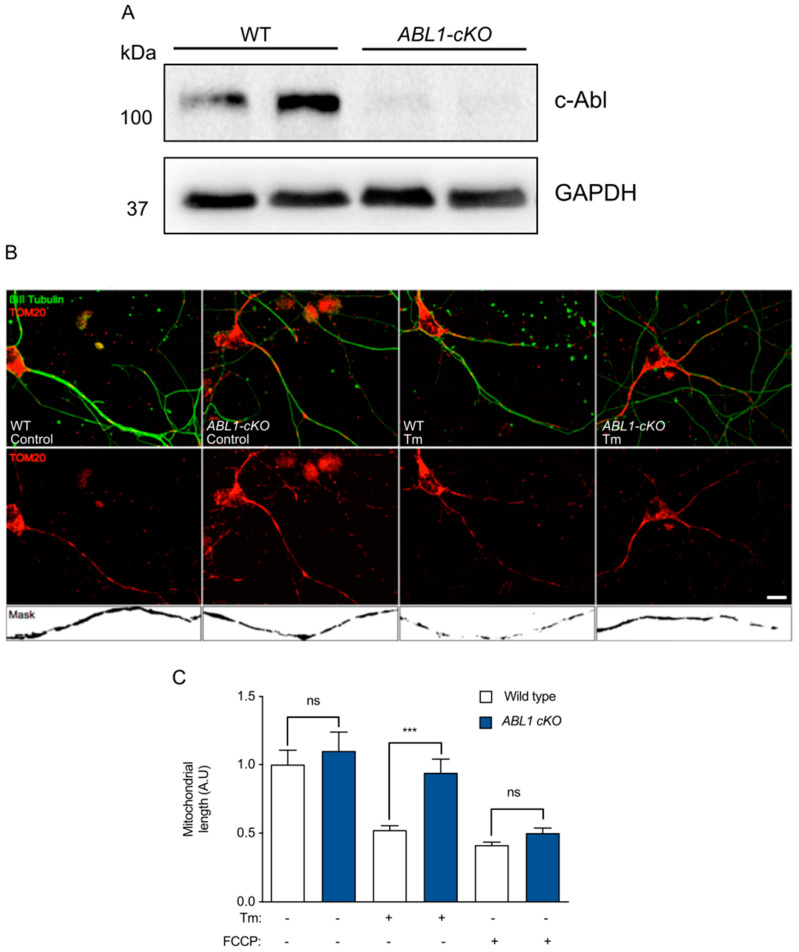
c-Abl regulates mitochondrial morphology in hippocampal primary neurons in response to ER stress. (**A**) c-Abl expression in primary neurons from WT and c-Abl-deficient (ABL1-cKO) mice. (**B**) Representative images of WT and ABL1-cKO primary hippocampal neurons treated with Tm for 8 h and immunostained for anti βIII tubulin (green) as the neuronal marker and TOM20 (red) as the mitochondrial marker. Scale bar: 5 μm. (**C**) Quantification of mitochondrial length from neuronal processes from experiments performed in (**B**). One-way ANOVA with Bonferroni post-test. Results are presented as mean ± SEM. ns: not significant, *p* *** < 0.001.

**Figure 3 antioxidants-12-02007-f003:**
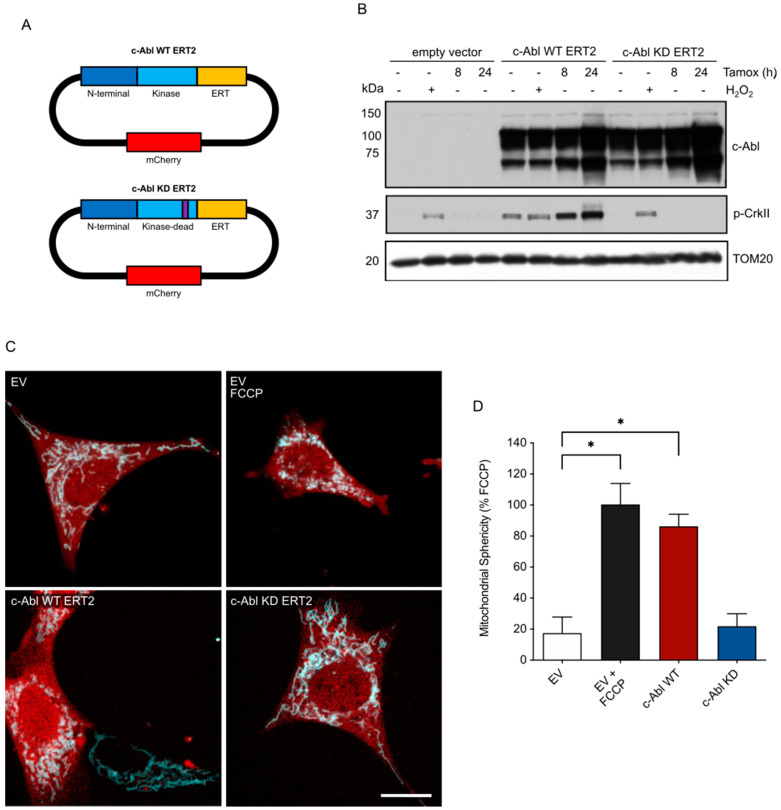
c-Abl activity induces mitochondrial fragmentation. (**A**) Generation of a tamoxifen-inducible system for constitutively active c-Abl (c-Abl WT ERT2) or kinase-dead (c-Abl KD ERT2) variants. (**B**) Activity of c-Abl variants in MEFs after ER stress was checked by immunoblot of c-Abl and a well-known substrate of c-Abl, p-Crk II. (**C**) Confocal microscopy of MEF cells overexpressing mitochondrial Cerulean-OMM (cyan) and the mCherry (red) empty vector (EV), c-Abl WT ERT2, or c-Abl KD ERT2 system, revealing the mitochondrial morphology under stimulation with tamoxifen. Scale bar: 20 μm. (**D**) Quantification of mitochondrial sphericity expressed as the percentage of sphericity relative to FCCP treatment in MEFs expressing the EV. One-way ANOVA with Bonferroni post-test. Results are presented as mean ± SEM. *p* * < 0.05.

**Figure 4 antioxidants-12-02007-f004:**
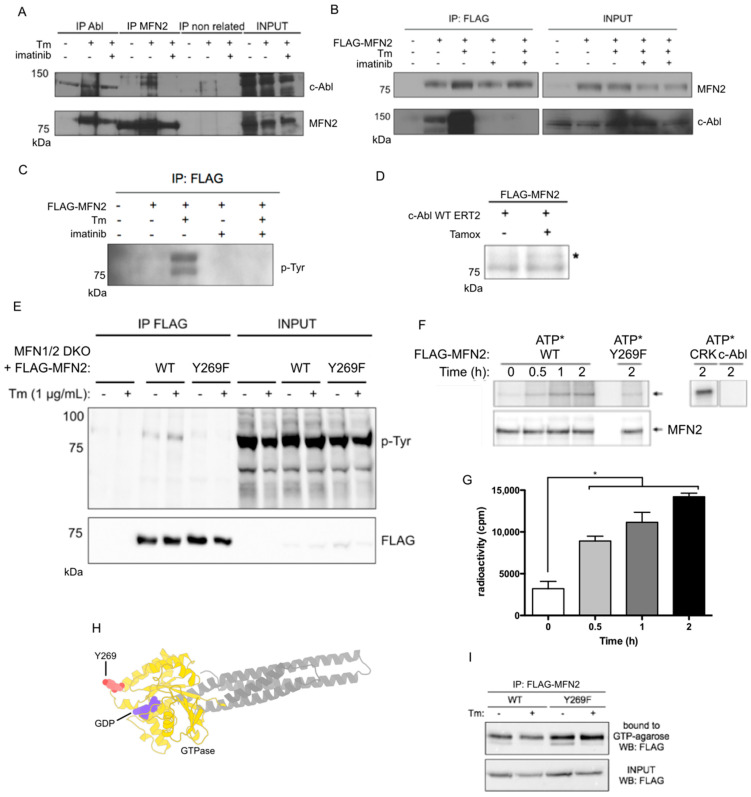
Activated c-Abl phosphorylates MFN2 in Y269. (**A**) Co-immunoprecipitation assay against endogenous levels for MFN2 and c-Abl in Tm or Ima treatments in MEF cells. (**B**) Co-immunoprecipitation assay for MEF cells over-expressing FLAG-MFN2 using agarose-anti-FLAG beads and detecting c-Abl in Tm or Ima treatments in MEF cells. (**C**) Phospho-tyrosine immunodetection against immunoprecipitated FLAG-MFN2 after Tm or Ima treatments in MEF cells. (**D**) In vitro phosphorylation with P32 orthophosphoric acid in living MEF cells and immunoprecipitation of FLAG-Mfn2 after tamoxifen (tamox)-stimulated c-Abl WT ERT2. (**E**) Phospho-tyrosine detection for MEF cells over-expressing either FLAG-MFN2 WT or FLAG-MFN2 Y269F variants in response to Tm. (**F**) In vitro phosphorylation of immunoprecipitated MFN2-FLAG, P32 gamma ATP (2.5 μCi), and recombinant c-Abl. (**G**) Scintillation counting of radiolabeled MFN2-FLAG after indicated times. (**H**) Modelling of Y269 residue in the MFN2 GTPase domain. (**I**) FLAG-MFN2 was immunoprecipitated and then incubated with GTP-agarose beads for 60 min at 30 °C. The bound proteins were analyzed via immunoblotting using FLAG antibody. One-way ANOVA with Bonferroni post-test. Results are presented as mean ± SEM. *p* * < 0.05.

**Figure 5 antioxidants-12-02007-f005:**
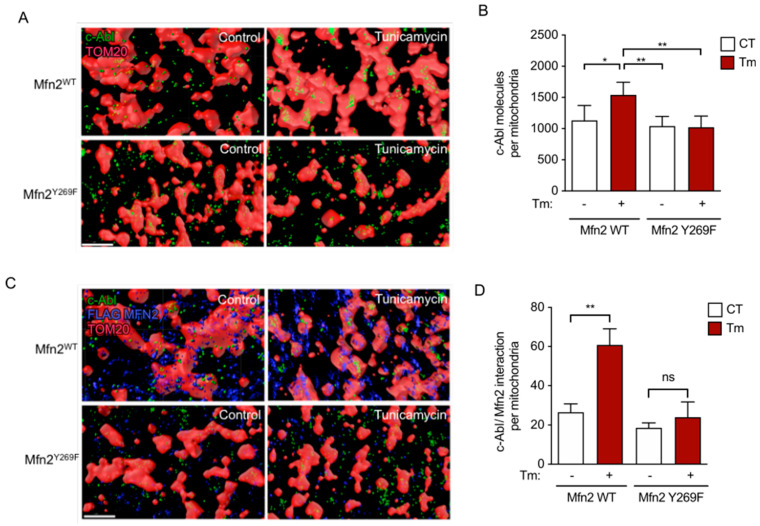
Activated c-Abl colocalizes with mitochondria dependent on MFN2 interaction. (**A**) Super-resolution microscopy revealing the c-Abl (green) localization in TOM20 (red)-stained FLAG-MFN2 WT or FLAG-MFN2 Y269F MEFs. Scale bar: 1 μm. (**B**) Quantification of c-Abl molecules as observed in (**A**). (**C**) Super-resolution microscopy of c-Abl (green) colocalization with FLAG-MFN2 (blue) and TOM20 (red) in FLAG-MFN2 WT or FLAG-MFN2 Y269F MEFs. Scale bar: 1 μm. (**D**) Quantification of c-Abl colocalizing with FLAG-MFN2. One-way ANOVA with Bonferroni post-test. Results are presented as mean ± SEM. ns: not significant, *p* * < 0.05, *p* ** < 0.01.

**Figure 6 antioxidants-12-02007-f006:**
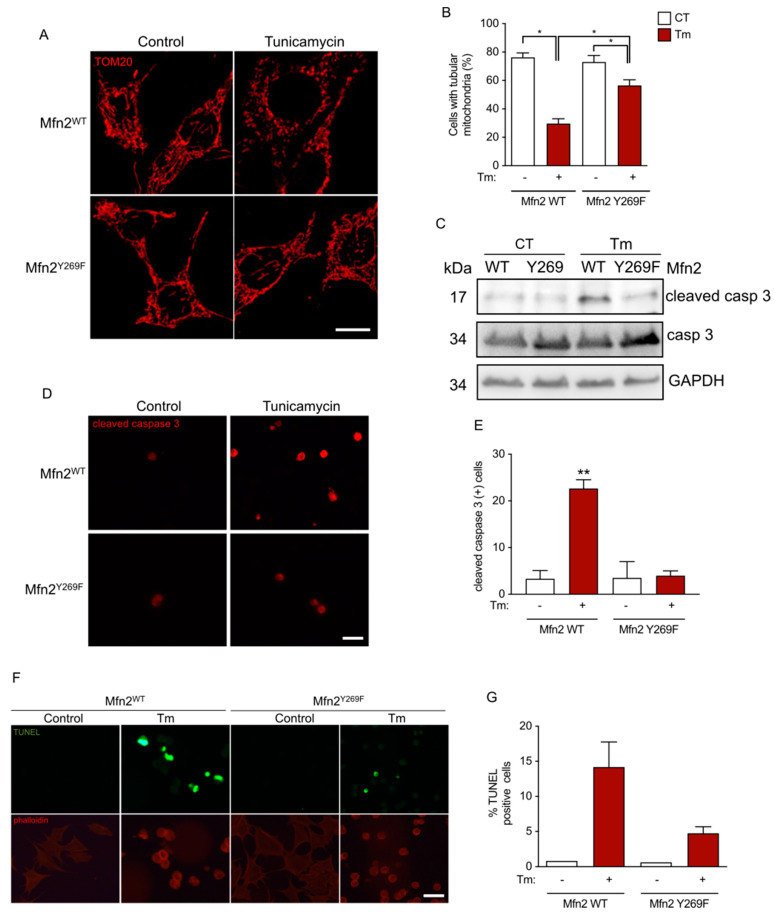
Mfn2 collaborates in the mitochondrial fragmentation and the apoptotic response in a Y269 residue-dependent manner. (**A**) Mfn2WT- and Mfn2Y269F-expressing cells were stained for TOM20 (red) and treated with vehicle (control) or exposed to Tm for 10 h. Scale bar: 20 μm. (**B**) The percentage of cells exhibiting tubular mitochondria in (**A**) was quantified. (**C**) Western blot against cleaved caspase 3 from Mfn2WT- and Mfn2Y269F-expressing cells exposed to Tm for 18 h. (**D**) Mfn2WT- and Mfn2Y269F-expressing cells were stained for cleaved caspase 3 (red) and treated with vehicle (control) or exposed to Tm for 18 h. Scale bar: 40 μm. (**E**) The number of cells per field exhibiting tubular mitochondria in (**D**) was quantified. (**F**) Mfn2WT- and Mfn2Y269F-expressing cells were developed for TUNEL staining (green) and phalloidin (red), and treated with vehicle (control) or exposed to Tm for 18 h. Scale bar: 40 μm. (**G**) The percentage of cells positive for TUNEL in (**F**) was quantified. One-way ANOVA with Bonferroni post-test. Results are presented as mean ± SEM. *p* * < 0.05, *p* ** < 0.01.

**Figure 7 antioxidants-12-02007-f007:**
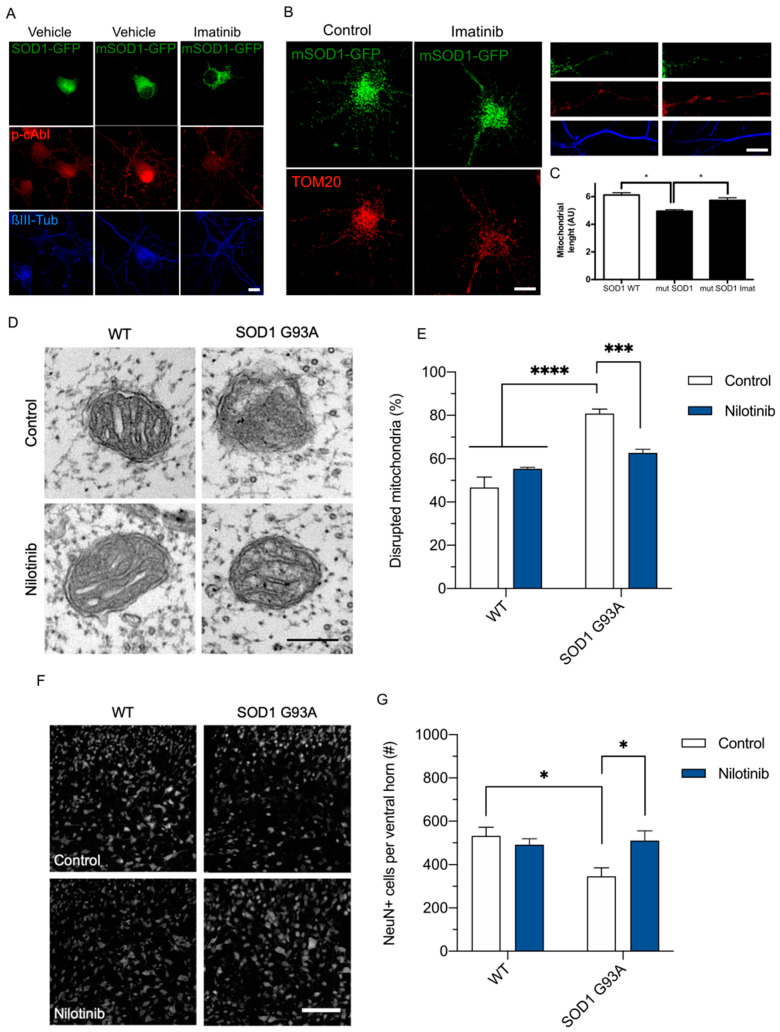
c-Abl regulates mitochondrial status in ALS models. (**A**) Confocal microscopy of primary hippocampal neurons overexpressing SOD1 WT fused to GFP (SOD1-GFP) or SOD1 G85R fused to GFP (mSOD1-GFP) in green, and stained for p-c-Abl (red) and βIII tubulin (blue), treated with vehicle or imatinib. Scale bar: 5 μm. (**B**) Confocal microscopy of primary hippocampal neurons overexpressing SOD1-GFP or mSOD1-GFP in green, and stained for TOM20 (red) both in soma (left side of the panel) and processes (right side of the panel) stained for and βIII tubulin (blue). Scale bar: 5 μm. (**C**) Mitochondrial length in neuronal processes as in (**B**). (**D**) Representative TEM images of mitochondria in the sciatic nerve from WT control-fed (n = 4) and nilotinib-fed (n = 4) mice and SOD1 G93A control-fed (n = 5) and nilotinib-fed (n = 5) mice. Scale bar = 500 nm. (**E**) Percentage of swollen mitochondria characterized by a disrupted external mitochondrial membrane with fragmented or swollen cristae/matrix in the sciatic nerve from (**D**). (**F**) Representative fluorescence images of NeuN in the ventral horn of the lumbar spinal cord in WT control-fed (n = 6) and nilotinib-fed (n = 5) mice, and SOD1 G93A control-fed (n = 5) and nilotinib-fed (n = 5) mice. Scale bar: 150 μm. (**G**) Graph shows the number of NeuN positive cells in 300,000 μm^2^ from (**F**). Two-way ANOVA with Tukey’s multiple comparisons test. Results are presented as mean ± SEM. *p* * < 0.05, *p* *** < 0.001, *p* **** < 0.0001.

**Table 1 antioxidants-12-02007-t001:** Sites in mitochondrial proteins predicted to undergo phosphorylation by c-Abl.

Protein ID *	Protein Name	Residue	Sequence −7/+7	Kinase-Dependent Predicted Value **	Hydrophobicity-Dependent Predicted Value ***	Score
O95140	MFN2	Y269	ASASEPEYMEEVRRQ	11.267	−1.527	17.205
Q8IWA4	MFN1	Y248	ASASEPEYMEDVRRQ	10.706	−1.527	16.348
Q9BXK5	B2L13	Y213	LESEEEEYPGITAED	10.014	−1.333	13.349
O75323	NIPS2	Y187	PRSGPNIYELRSYQL	11.430	−1.120	12.802
Q5THJ4	VP13D	Y2873	TNLEHQIYARAEVKT	12.250	−1.040	12.740
Q8IWA4	MFN1	Y40	SHFVEATYKNPELDR	10.319	−1.213	12.517
O00429	DNM1L	Y266	TDSIRDEYAFLQKKY	9.849	−1.220	12.016
Q5THJ4	VP13D	Y768	TQFSDDEYKTPLATP	9.884	−1.180	11.663
Q969Q5	RAB24	Y70	DTAGSERYEAMSRIY	10.385	−1.093	11.351
O00429	DNM1L	Y101	LHTKNKLYTDFDEIR	9.714	−1.153	11.200
Q5THJ4	VP13D	Y1189	GMANREKYGRKIATA	11.108	−0.987	10.964
Q9BUR5	MIC26	Y43	KVDELSLYSVPEGQS	13.270	−0.667	8.851
Q96HS1	PGAM5	Y224	ARQEEDSYEIFICHA	12.029	−0.680	8.180
P57105	SYJ2B	Y43	VSNDSGIYVSRIKEN	10.196	−0.767	7.820
Q6UXV4	MIC27	Y44	KPEQLPIYTAPPLQS	10.986	−0.700	7.690
Q9BPW8	NIPS1	Y262	GWDENVYYTVPLVRH	14.185	−0.533	7.561
Q9BPW8	NIPS1	Y87	KPEYLDAYNSLTEAV	11.845	−0.600	7.107
Q16611	BAK	Y108	QPTAENAYEYFTKIA	9.754	−0.720	7.023
O43236	SEPT4	Y318	EHFGIKIYQFPDCDS	10.827	−0.593	6.420
Q14318	FKBP8	Y187	GPQGRSPYIPPHAAL	9.852	−0.627	6.177
Q8NAN2	MIGA1	Y152	KGSQVCNYANGGLFS	12.228	−0.333	4.072
O43865	SAHH2	Y28	EIEDAEKYSFMATVT	9.989	−0.347	3.466
Q07817	B2CL1	Y120	HITPGTAYQSFEQVV	9.800	−0.167	1.637
O75323	NIPS2	Y264	GWEELVYYTVPLIQE	11.294	0.020	−0.226
O43236	SEPT4	Y228	CWKPVAEYIDQQFEQ	10.305	N/A	N/A
O43236	SEPT4	Y407	RETHYENYRAQCIQS	10.99	N/A	N/A
O43865	SAHH2	Y470	ALALIELYNAPEGRY	9.678	N/A	N/A
O60313	OPA1	Y637	THVIENIYLPAAQTM	9.659	N/A	N/A
O75323	NIPS2	Y89	KPECLEAYNKICQEV	9.56	N/A	N/A
O75323	NIPS2	Y231	FSQIGQLYMVHHLWA	9.553	N/A	N/A
Q5HYI7	MTX3	Y29	ESLVVMAYAKFSGAP	11.314	N/A	N/A
Q5THJ4	VP13D	Y3589	QDNRQLYYENFIYIA	9.867	N/A	N/A
Q5THJ4	VP13D	Y3861	LTGINVHYTQLATSH	10.98	N/A	N/A
Q5THJ4	VP13D	Y4369	NYAKSLYYEQQLMLR	10.067	N/A	N/A
Q6UXV4	MIC27	Y18	TMPAGLIYASVSVHA	10.63	N/A	N/A

* Protein ID in UniProt database. ** Value obtained after FASTA sequence analysis in GPS 5.0 software. *** Hydrophobicity value of the residue reported by PhosphoNET online tool. N/A: Not applicable.

## Data Availability

The data presented in this study are available on request from the corresponding author. The data are not publicly available due to the proprietary nature of the drug neurotinib and its associated patent WO2019/173761 A1, which restricts sharing the data without prior authorization from the corresponding author and a valid, justified request.

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
