# Peer review of "c-Abl Phosphorylates MFN2 to Regulate Mitochondrial Morphology in Cells under Endoplasmic Reticulum and Oxidative Stress, Impacting Cell Survival and Neurodegeneration"

_antioxidants, 2023, doi:10.3390/antiox12112007_

Round 1
Reviewer 1 Report
Comments and Suggestions for Authors
This manuscript focused on the role of phosphorylation of MFN2 by c-Abl under ER and oxidative stress. The authors provided the observations of mitochondrial morphology, MFN2 translocation and cell survival that related to MFN2 phosphorylation. In addition, the authors demonstrated that c-Abl inhibitor delays symptoms onset in ALS mouse model. The manuscript provided evidence of MFN2 phosphorylated by c-Abl and potential link to ALS. It will be interesting to readers in mitochondria dynamics and neurodegenerative disorders research fields.
Major Concerns
l It is interesting that the manuscript demonstrated that MFN2 could be phosphorylated by c-Abl under ER stress. However, the molecular mechanism of phosphorylation of MFN2 on Y269 by c-Abl could be elaborated. For example, phosphor-mimetic MFN2 Y269D/E could be used for examining c-Abl interaction with MFN2. Examining whether Y269D/E had dominated effect on mitochondria morphology is the other way to demonstrate the role of MFN2 Y269 phosphorylation.
l Specific antibody against phosphorylation at MFN2 Y269 will also aid to address ER stress issue.
l
Minor concerns:
l The ALS animal model results have less correlation to MFN2 phosphorylation. It is the reviewer’s suggestion that the authors should either provide the MFN2 Y269 phosphorylation condition results to make it comprehensive or better conjugation.
l The effects of MFN2 phosphorylation by c-Abl on mitochondrial activity will aid readers to appreciate the manuscript.
Comments on the Quality of English LanguageMinor editing is suggested.
Author Response
First we would like to thank to the reviewer 1 for the positive feedback on our manuscript. We have made changes asked to improve the manuscript based on reviewer 1 comments and suggestions
Major Concerns:
1. The reviewer asks for an elaboration of the molecular mechanism behind MFN2 phosphorylation by c-Abl, suggesting the use of phosphor-mimetic MFN2 mutants (Y269D/E) to examine the interaction with c-Abl and its effects on mitochondrial morphology.
We appreciate your valuable feedback and suggestions. We have carefully considered your major concern regarding the molecular mechanism behind MFN2 phosphorylation by c-Abl and the use of phosphor-mimetic mutants, specifically Y269D/E, to examine their interaction and effects on mitochondrial morphology.
Firstly, we would like to acknowledge your suggestion to employ Y269D/E mutants to explore the impact of phosphorylation on mitochondrial morphology. However, at this point, we do not have phosphor-mimetic mutants for MFN2 at residue Y269. While we agree that this strategy could provide valuable insights into the effects of phosphorylation, it is important to highlight that we can examine the impact of the Y269 residue itself on the interaction between c-Abl and MFN2. We believe that this alternative approach will still contribute to our understanding of the molecular mechanism, even in the absence of phosphor-mimetic mutants for Y269. We discussed this point in the manuscript by adding the following sentence ‘The use of other MFN2 mutants such as phosphomimetic MFN2 Y269E could be included in future experiments to explore more deeply the contribution on mitochondrial morphology, however our approach confirm that the Y269 participates in c-Abl MNF2 interaction.’
2. The reviewer recommends the use of a specific antibody against phosphorylation at MFN2 Y269 to address the ER stress issue.
- We acknowledge the importance of this suggestion. Currently there is no commercial antibody against MFN2 phosphorylated in Y269 and unfortunately, despite our attempts, we were unable to generate a specific antibody targeting MFN2 phosphorylation at Y269, and the results were not as desired.
Minor Concerns:
1. The reviewer finds a lack of correlation between the ALS animal model results and MFN2 phosphorylation. They suggest providing MFN2 Y269 phosphorylation condition results or better conjugation to make it comprehensive.
- We appreciate the reviewer's feedback. We would like to clarify that, unfortunately, we do not have access to an antibody specific to MFN2 phosphorylation at Y269.
However, we would like to highlight that our in vitro results show an increased fragmented phenotype of mitochondria upon ER stress induction, and c-Abl inhibition or ablation restores mitochondrial phenotype (Figures 1, 2, 3). Additionally, we have observed decreased tyrosine phosphorylation in MFN2 both after c-Abl inhibition and resistant to the Y269F mutant (Figure 4) and our in vivo results indicate increased disrupted mitochondrial phenotype and neuronal loss in an ALS model, which is dependent on c-Abl activity (Figure 7). These findings correlate with our in vitro observations regarding c-Abl activity's impact on mitochondrial function and cell death induction. To emphasize these correlations, we added the following paragraph to the discussion:
“Overall, our study provides a view into the complex relationship between c-Abl kinase and mitochondrial dynamic mediated by MFN2 phosphorylation at Y269 during ER stress, mainly in the context of ALS. The correlation between in vitro and in vivo results reinforces the significance of c-Abl activity in mediating mitochondrial responses to stress conditions and the use of c-Abl inhibitors for targeting mitochondrial dysfunction in ALS.”
2. The reviewer suggests investigating the effects of MFN2 phosphorylation by c-Abl on mitochondrial activity to enhance the manuscript's value.
- We agree with this suggestion. To emphasize the impact of MFN2 phosphorylation by c-Abl on mitochondrial activity we measured OCR and ECAR in MFN WT or Y269F MEF and included these results in Figure S4. The following paragraph was added to describe this data:
“Mitochondrial dynamics have the potential to influence mitochondrial function. In our observations, we found that ER stress did not affect the oxygen consumption rate (OCR) response (Figure S4A and S4B). However, it did lead to a reduction in the extracellular acidification rate (ECAR) in MFN2 WT MEFs (Figure S4C), indicating that the cells were still capable of oxygen consumption, but with altered glycolytic activity. In the case of the MFN Y269F mutant MEFs, treatment with tunicamycin did not produce changes in either OCR (Figure S4A and S4B) or ECAR (Figure S4C), suggesting that these cells could rely on both oxidative phosphorylation and glycolysis even when subjected to ER stress.”
Reviewer 2 Report
Comments and Suggestions for Authors
This paper is very well written and the thesis is well thought out. There doesn't seem to be any major problem.
There are a few small points to make about the figures. Throughout, there are some photos that do not have scale bars, and some that are not mentioned in the descriptions. Please include at least one in your photo. Also, please write the scale length.
Regarding scale, the size of the cells in the upper and lower rows of Figure 1 appears to be clearly different. If you have changed the magnification, please state it clearly. Regarding Figure 1D, it is a graph of mitochondrial length, but is this a statistical process from Figure 1E? If so, the picture should come first as a way to present the diagram.
Regarding WB, I think each Y axis is probably kDa, but please provide.
Figure 5 is shown again on page 31. What is this?
Author Response
First we would like to thank to the reviewer 2 for the positive feedback on our manuscript. We have made changes asked to improve the manuscript based on reviewer 2 comments and suggestions.
- The reviewer notes that some photos lack scale bars and mentions in the descriptions. They request the inclusion of at least one scale bar in the figures and providing scale length.
- We ensured that all figures include scale bars and provide scale lengths for clarity and accuracy.
- The reviewer points out a discrepancy in cell size in Figure 1 and suggests stating if there was a change in magnification.
- Effectively, upper and lower rows of Figure 1A have different magnification, we modified the lower row to match the magnification of the upper one.
- Regarding Figure 1D (Mitochondrial length), the reviewer questions if it's a statistical process from Figure 1E and suggests presenting the picture before the diagram.
- We appreciate the reviewer's observation. To clarify, Figure 1D represents a statistical analysis derived from the data presented in Figure 1C (mitochondrial length in neurons), while Figure 1F is a statistical analysis derived from Figure 1E (Sphericity index from mitochondria in MEF).
- The reviewer asks for clarification on the units (kDa) of the Y-axis in the Western Blot (WB) figures.
- We explicitly mention that the Y-axis in the WB figures represents kDa for clarity.
- The reviewer inquires about the appearance of Figure 5 on page 31 and seeks clarification.
- We reviewed page 31 to ensure that Figure 5 is correctly placed and that any discrepancies are addressed or corrected.
Round 2
Reviewer 2 Report
Comments and Suggestions for Authors
The authors have sincerely and appropriately revised my previous comments. As a reviewer, I have no further objections to this paper.